# Melanogenic Inhibition and Toxicity Assessment of Flavokawain A and B on B16/F10 Melanoma Cells and Zebrafish (*Danio rerio*)

**DOI:** 10.3390/molecules25153403

**Published:** 2020-07-28

**Authors:** Nurshafika Mohd Sakeh, Nurliyana Najwa Md Razip, Farah Idayu Mohd Ma’in, Mohammad Nazri Abdul Bahari, Naimah Latif, Muhammad Nadeem Akhtar, Zetty Norhana Balia Yusof, Syahida Ahmad

**Affiliations:** 1Department of Biochemistry, Faculty of Biotechnology and Biomolecular Sciences, Universiti Putra Malaysia, UPM Serdang, Serdang 43400, Selangor, Malaysia; shafikams@gmail.com (N.M.S.); najwabiochem@gmail.com (N.N.M.R.); farahidayu1991@gmail.com (F.I.M.M.); nazrischolar@gmail.com (M.N.A.B.); naimah_latif@yahoo.com (N.L.); zettynorhana@upm.edu.my (Z.N.B.Y.); 2Faculty of Industrial Sciences & Technology, Universiti Malaysia Pahang, Lebuhraya Tun Razak, Gambang, Kuantan 26300, Pahang, Malaysia; nadeem@ump.edu.my

**Keywords:** α-MSH, B16/F10 melanoma, chalcone, flavokawain, melanogenesis, zebrafish

## Abstract

Excessive production of melanin implicates hyperpigmentation disorders. Flavokawain A (FLA) and flavokawain B (FLB) have been reported with anti-melanogenic activity, but their melanogenic inhibition and toxicity effects on the vertebrate model of zebrafish are still unknown. In the present study, cytotoxic as well as melanogenic effects of FLA and FLB on cellular melanin content and tyrosinase activity were evaluated in α-MSH-induced B16/F10 cells. Master regulator of microphthalmia-associated transcription factor (*Mitf*) and the other downstream melanogenic-related genes were verified via quantitative real time PCR (qPCR). Toxicity assessment and melanogenesis inhibition on zebrafish model was further observed. FLA and FLB significantly reduced the specific cellular melanin content by 4.3-fold and 9.6-fold decrement, respectively in α-MSH-induced B16/F10 cells. Concomitantly, FLA significantly reduced the specific cellular tyrosinase activity by 7-fold whilst FLB by 9-fold. The decrement of melanin production and tyrosinase activity were correlated with the mRNA suppression of *Mitf* which in turn down-regulate *Tyr*, *Trp*-1 and *Trp*-2. FLA and FLB exhibited non-toxic effects on the zebrafish model at 25 and 6.25 µM, respectively. Further experiments on the zebrafish model demonstrated successful phenotype-based depigmenting activity of FLA and FLB under induced melanogenesis. To sum up, our findings provide an important first key step for both of the chalcone derivatives to be further studied and developed as potent depigmenting agents.

## 1. Introduction

Impairment of mechanisms regulating melanogenesis may cause overproduction of melanin, resulting in various hyperpigmentation disorders. In addition to excessive exposure to ultraviolet radiation (UVR), post-inflammatory melanoderma occurring in healing after inflammation [1,2,3] and involuntary activation of alpha-melanocyte stimulating hormone (α-MSH) receptors by adrenocorticotropic hormone (ACTH) associated to Addison’s disease may result in hyperpigmentation [3,4]. Melanocortin 1 receptor (MC1R), a G-protein-coupled receptor located in the membrane of melanocytes is responsible in regulating skin pigmentation [5]. Both α-MSH and ACTH are melanocortins derived from proteolytic cleavage of pro-opiomelanocortin (POMC) which are endogenous agonists for MC1R [3,6]. Activation of MC1R triggers signaling cascade of cyclic adenosine monophosphate (cAMP) pathway through stimulation of adenylate cyclase which leads to activation of protein kinase A (PKA). Activated PKA further phosphorylates cAMP-responsive element binding (CREB) transcription factor which in turn, promotes transcriptional activation of microphtahlmia-associated transcription factor (*Mitf*) [7]. *Mitf* gene plays important role as master regulator of melanogenesis. It is essentially associated with the regulation of three other melanogenic-related genes namely tyrosinase (*Tyr*), tyrosinase-related protein 1 (*Trp*-1) and tyrosinase-related protein 2 (*Trp*-2) via E-box of *Mitf* basic-helix-loop-helix-leucine-zipper (bHLH-LZ) [8,9]. Melanin, represented as eumelanin, pheomelanin, neuromelanin, and mixed melanin are the end products of complex metabolic steps that initiated from amino acid l-tyrosine precursor [10]. Tyrosinase (EC 1.14.18.1) plays a crucial role as key regulatory as well as rate-limiting enzyme in converting l-tyrosine into substrates that finally form melanin. Dopachrome tautomerase which is abbreviated as *Trp*-2 functions in isomerization of intermediate 5,6-dihydroxyindole (DHI) to 5,6-dihydroxyindole-2-carboxylic acid (DHICA) [11]. Subsequently, *Trp*-1 also known as DHICA oxidase, converts DHICA to indole-5,6-quinone-carboxylic acid, catalyzing eumelanogenesis [12,13]. 

Abnormalities of skin tone due to pigmenting disorders results in formidable emotional and social problems [14,15]. The earliest established depigmenting agent, hydroquinone demonstrated strong depigmentation effect [16,17], but also raised the concerns of carcinogenicity and mutagenicity [18], and was known to permanently caused damage to melanocytes [19]. Kojic acid and arbutin which are structurally similar to l-DOPA and l-tyrosine, were later discovered as tyrosinase inhibitors by chelating copper in tyrosinase’s active site [20,21,22]. However, only arbutin is widely being used by cosmetic consumers since kojic acid was reported to promote hypersensitivity in skin [23], hepatocarcinogenesis [24] and mutagenic [25]. Hence, the elevation of public concern on drugs safety has augment the development of effective yet harmless depigmenting agents.

Toxicology assessment requires large, laborious, time consuming and expensive preclinical trials utilizing in vivo models such as mice and guinea pigs. On the contrary, zebrafish (*Danio rerio*) model has been identified as an ideal model organism for large screening due to their smaller in size, easy to maintain, readily reproducible in laboratories, externally and rapidly developing transparent embryos permitting live observations with high throughput chemical screening [26]. The zebrafish model has also been extensively used as a resort to toxicity assessment of drug compounds [27,28,29] and melanogenesis studies since the lower vertebrate is highly comparable to human melanogenesis [30]. The presence of melanin pigments on surface of zebrafish larvae allow ready observation in testing percutaneous effect of depigmenting agents [31]. Moreover, screening of embryotoxicity of drugs or small molecules involves rapid and simple procedure whereby compounds diluted in embryo media is readily absorbed through skin and gills during early stage of embryo [32,33]. 

Flavokawain A (FLA) and flavokawain B (FLB) were naturally occurring kava chalcones first extracted from Kava (*Piper methysticum*), a plant grown in the Pacific Islands [34]. FLA and FLB were then reported to have significant effects as anti-cancer [35,36], antinociceptive [37] and anti-inflammatory [38,39]. FLA and FLB have been reported to have anti-tyrosinase activity in our previous study [40]. However, the depigmenting mechanisms on B16/F10 cells have only been reported on FLB, but not on FLA. Thus, in the present study the anti-melanogenic activity of FLA and FLB were evaluated on α-MSH-induced B16/F10 melanoma cells by validating the expression level of *Mitf*, *Tyr*, *Trp*-1 and *Trp*-2. Toxicity assessments (embryotoxicity and teratogenicity) and in vivo phenotype-based depigmenting effects were also performed on zebrafish model to further elucidate FLA and FLB as potent yet safe depigmenting agents.

## 2. Results

### 2.1. Cytotoxic Effect of Chalcone Derivatives on B16/F10 Cells

The toxicity effects of FLA and FLB on α-MSH-induced B16/F10 cells are illustrated as in Figure 1. In the present study, a commercialized depigmenting agent (namely arbutin) was also evaluated for its cytotoxic effect. Both FLA and FLB showed no significant toxicity effect towards induced B16/F10 cells after incubation for 72 h up to concentration 25 µM with cell viability of 86.66% ± 1.78 (FLA) and 85.81% ± 0.78 (FLB). At highest concentration of 50 µM, FLA and FLB showed significant cytotoxic effect with cell viability decreased to 85.26% ± 1.35 and 80.41% ± 1.75, respectively. The results demonstrated that FLA showed lesser cytotoxic effect than FLB. On the other hand, arbutin (positive control) showed no significant cytotoxic effect with more than 90% of cell viability at 50 µM. Control was set as α-MSH-induced B16/F10 in 0.1% DMSO which serve as carrier solvent of compounds.

### 2.2. Inhibitory Effect of Chalcone Derivatives on Cellular Melanin Content

Melanin is the final product of a complete melanin biosynthesis pathway. A potent depigmenting agent is described with respect of decrement in melanin formation. The depigmenting effect of both FLA and FLB on α-MSH-induced B16/F10 cells is described as in Figure 2. Induced melanogenesis by α-MSH results in significant elevation of cellular melanin production. In order to study the decrement production of cellular melanin content, induced B16/F10 cells were treated with FLA and FLB by 4-fold serial dilution of compounds (25, 6.25 and 1.56 µM). Results demonstrated high extent of depigmenting effect of FLA (0.84 ± 0.06 µg melanin/ µg protein) and FLB (0.38 ± 0.03 µg melanin/ µg protein) at highest concentration of 25 µM. In the absence of compounds, α-MSH-induced B16/F10 cells produced 3.75 ± 0.05 µg melanin/ µg protein. It was also found that both compounds showed inhibition towards cellular melanin production in dose-dependent manner. Interestingly, cellular melanin content of B16/F10 cells were reduced albeit at the lowest concentration of 1.56 µM. 

### 2.3. Inhibitory Effect of Chalcone Derivatives on Cellular Tyrosinase Activity

Tyrosinase plays a major role in the initial biosynthesis reaction of melanogenesis. Melanogenic inhibition targeting on tyrosinase has been a major interest in developing new depigmenting agent. The melanogenic effect of chalcone derivatives was evaluated on cellular tyrosinase activity of α-MSH-induced B16/F10 cells (Figure 3). In the study, enzyme activity was measured by using l-DOPA as enzyme substrate while cells were treated in 4-fold serial dilution of compounds dosages descending from 25 to 1.56 µM. The induced tyrosinase activity of cells was observed to decline tremendously with the treatment of FLA (1.23 ± 0.04 µU/µg protein) and FLB (0.96 ± 0.02 µU/µg protein) at highest concentration of 25 µM. In the absence of compounds, α-MSH-induced B16/F10 cells exhibited 8.64 ± 0.15 µU/µg protein of tyrosinase activity. It was also demonstrated that enzyme inhibitory activity of FLA and FLB exhibited a dose-dependent manner in which significant inhibition was evident up to 6.25 µM. 

### 2.4. Down-Regulation Effect of Chalcone Derivatives on Tyr, Trp-1, Trp-2 and Mitf Genes Expression in B16/F10 Cells

*Mitf* as well as *Tyr*, *Trp*-1, and *Trp*-2 genes are well-studied to regulate majorly in melanogenesis. The effect of FLA and FLB on mechanism of melanin biosynthesis pathway was elucidated on *Tyr*, *Trp*-1, *Trp*-2 and *Mitf* genes expression (Figure 4). In the present study, B16/F10 cells were exposed to various concentrations of chalcone derivatives (1.56, 6.25 and 25 µM). Induced expression of *Mitf*, *Tyr*, *Trp*-1 and *Trp*-2 genes in B16/F10 cells was observed after treatment with α-MSH for 72 h. Interestingly, a dose-dependent manner of gene suppression was observed on *Tyr*, *Trp*-1, *Trp*-2 and *Mitf* genes after treatment with FLA and FLB. At highest concentration of 25 µM, both FLA and FLB showed suppression on *Tyr* gene with 0.20 ± 0.01 and 0.15 ± 0.04-fold expression respectively. Suppression on the master regulator of *Mitf* gene was also found significant on both FLA and FLB at highest concentration of 25 µM with 0.41 ± 0.07 and 0.32 ± 0.03-fold expression, respectively. The other melanogenic-related genes of *Trp*-1 and *Trp*-2 demonstrated a similar pattern of gene suppression by both FLA and FLB. However, the genes were only significantly downregulated up to 6.25 µM by FLB. Arbutin at 50 µM was set as positive control and showed significant downregulation on all genes. In the assay, induced group (control) was set at 1.00-fold expression and the expression of all genes of interest was normalized in the reference of endogenous genes transcript amount (*GAPDH* and *ß-actin*).

### 2.5. Zebrafish Toxicity Assessment

3-isobutyl-1-methylxanthine (IBMX), a chemical compound was used in the experiment to increase melanin production regulated by the same pathway as α-MSH [41,42]. Figure 5 showed the survival rates of zebrafish post-treated with 10-fold serially diluted FLA and FLB (0.78–50 µM) in 0.1% DMSO with the presence of IBMX inducer at 100 µM. The induced group which consisted of zebrafish embryo in embryo media with 0.1% DMSO and IBMX (100 µM) demonstrated 100% survival rate until the end of testing period. Twelve embryos were selected and incubated at 9–144 hours-post-fertilization (hpf). The survival rate of zebrafish was concluded based on the presence of heart beat visual and the absence of teratogenic effect. Based on the heart beat visual, the findings demonstrated high survival rate of zebrafish with higher than 80% when treated with FLA from 0.78–25 µM while FLB from 0.78–12.5 µM. However, teratogenic effect has been detected on zebrafish treated with FLB at 12.5 µM. Hence, the optimal concentrations of FLA and FLB for melanogenic inhibitory tests were determined at 25 µM and 6.25 µM, respectively. The LC_50_ values calculated for FLA and FLB were at 45 µM and 25 µM, respectively.

### 2.6. Zebrafish Phenotype-Based of Melanogenic Inhibition by FLA and FLB

Synchronized embryos collected at 9 hpf were treated with FLA and FLB at 25 µM and 6.25 µM, respectively (Figure 6). Uninduced zebrafish group demonstrated normal melanogenesis of zebrafish in embryo media with the presence of 0.1% DMSO. Induced zebrafish group treated with IBMX at 100 µM exhibited excessive production of melanin. In the presence of phenylthiourea (PTU) at 200 µM (positive control), melanin production of zebrafish was successfully inhibited from 9 to 96 hpf. After PTU-washed off and treatment with IBMX at 96 hpf, the zebrafish pigmentation of positive control was found recovered at 144 hpf. Similarly, pigmentation of zebrafish was inhibited with PTU (200 µM) for FLA and FLB treatment groups from 9 to 96 hpf. The zebrafish depigmenting of both groups was revoked by substitution of PTU with IBMX (100 µM) melanin inducer. The melanin recovery exhibited by zebrafish groups treated with FLA and FLB after PTU-washed off and IBMX-induced at 96 hpf, was lower as compared to both positive and negative controls. The results indicate the ability of both compounds to inhibit and/or reduce the zebrafish melanogenesis.

## 3. Discussion

Researchers have been addressing tyrosinase inhibition as crucial target to reduce excessive production of melanin which causes hyperpigmentation [43,44]. Flavonoids are a group of most studied polyphenols with the ability of chelating copper ions in tyrosinase’s active site [45,46]. According to the new classification of polyphenols (Phenol-Explorer Version 3.6 database, www.phenol-explorer.eu), chalcones are considered as a subclass of flavonoids [47]. In addition to the anti-tyrosinase property [40,48,49], chalcones and their derivatives have been reported to possess various biological and pharmacological activities including anti-inflammatory, antipyretic, analgesic, bactericidal, insecticidal, anti-fungal anticancer and antioxidant [50]. Flavokawains used in the study are a class of chalcones found on kava plants, which were synthesized by base catalyzed Claisen–Schmidt condensation (Figure 7A). In our preliminary studies, anti-tyrosinase activity and structural activity relationship of FLA (IC_50_ = 14.26 ± 0.08 μM) and FLB (IC_50_ = 14.38 ± 0.12 μM) have been evaluated [40]. The findings are consistent with reports that suggested 4-subtituted methoxy group in B ring of FLA significantly contributes to better anti-tyrosinase activity of chalcones [48,51]. Tyrosinase is a copper-containing enzyme which plays a crucial role as rate-regulator of melanin synthesis pathway whereby it catalyzes the hydroxylation of l-tyrosine precursor to l-DOPA and further oxidized to dopaquinone [52]. It was postulated that the amount of tyrosinase activity, is equal to the amount of melanin synthesized [53]. Due to this vital fact, development of new whitening agent has been targeting tyrosinase in the aim to combat various melanogenic-related diseases. It was suggested that the prominent inhibitory activities of chalcone on tyrosinase were due to its resemblance structure with the substrates of tyrosinase, namely tyrosine and l-DOPA (Figure 7B).

In the early efforts to determine the potential of FLA and FLB as safe and reliable depigmenting agents, the cytotoxic effects of both compounds were characterized on α-MSH-induced B16/F10 melanoma cells. It was shown that FLA and FLB did not exert cytotoxic effects until at 25 μM which allow further characterization of depigmenting activity on B16/F10 cells. Interestingly the functional property of both FLA and FLB as depigmenting compounds were successfully elucidated through in vitro study on B16/F10 cells revealing reduced melanin production in a dose-dependent manner. Jeong et al. [54] has previously reported the melanogenesis inhibition activity of FLB and FLC, but the FLA was found to exhibit no melanin inhibitory activity. Although the cellular melanin content was tested similarly in MSH-activated B16 melanoma cells, the authors did not determine the cellular tyrosinase inhibition activity of FLA and FLB. In this study, further in vitro evaluation of melanogenic inhibition by FLA and FLB was carried out by evaluating the inhibitory activity on cellular tyrosinase. Treatment of FLA and FLB has clearly showed decrement of α-MSH-induced cellular tyrosinase activity in B16/F10 cells. Our findings were able to correlate the successful inhibition of cellular tyrosinase activity with the reduction of cellular melanin content. At transcriptional level, our findings showed that both FLA and FLB significantly suppressed the melanogenic master regulator of *Mitf* gene expression which in turn suggesting the down-regulation of other melanogenic-related genes including *Tyr*, *Trp*-1 and *Trp*-2 [55]. The successful suppression of *Mitf* determines the effectiveness of FLA and FLB as potent depigmenting agents since *Mitf* is essential in melanin-bearing pigment cells across species and tissue types [55,56]. The outcome has suggested the implication of FLA and FLB in mechanistic inhibition on cAMP/α-MSH mechanism pathway. Interestingly, both in vitro results suggested FLB as a better anti-melanogenic candidate as compared to FLA.

α-MSH is one of bioactive melanocortin family peptides which derived from larger precursor peptide of POMC that is present in melanocytes and keratinocytes [57]. It was well-known that the minute concentration of α-MSH interacts simultaneously through cAMP second messenger via MC1R, which will then promote melanogenesis [58]. Under normal circumstances, over-exposure of UVR may activate excessive amount of α-MSH which results in the induced melanogenesis [3,7]. However, the pigmentary hormone of α-MSH has been reported to interfere normal biological function causing increased metabolic rate and cardiovascular dynamics in vertebrate [59,60]. Extracellular stimulus of IBMX results in hyperpigmentation by inducing cAMP which activates similar pathway caused by α-MSH [41,61]. Hence, IBMX was used as melanogenic inducer agent in the zebrafish model to further evaluate the phenotype-based toxicity and depigmenting activity of FLA and FLB. FLA demonstrated no toxicity on zebrafish embryo at 25 µM, which was the same concentration when tested in B16/F10 cells. However, FLB was only found non-toxic at 6.25 µM when tested on zebrafish embryo. The toxicity effect of FLA and FLB on zebrafish model may be influenced by factors including absorption and exposure time [26]. The higher toxicity effect of FLB as compared to FLA was in line with other reports [62,63], and zebrafish has become a more powerful approach as toxicological model with high genetic homology to mammals [64]. Intriguingly, the melanogenic inhibition of both FLA and FLB was also found effective on zebrafish vertebrate model even though the concentration used for FLB treatment was 2-fold lower than tested in B16/F10 cells. Both compounds were able to inhibit the production of melanin in zebrafish in the presence of IBMX melanin inducer agent. 

In summary, FLA and FLB demonstrated no toxicity effect in B16/F10 cells and zebrafish at 25 and 6.25 µM, respectively. Our findings showed that both compounds were also efficient in inhibiting melanogenesis up to molecular level by suppressing the expression patterns of master regulator *Mitf* which concomitantly decreased the expression of other melanogenic-related genes including *Tyr*, *Trp*-1 and *Trp*-2. Elucidation of the downstream regulations by the depigmenting agents as well as its elaborated toxicity assessment are important as the first step in the development of whitening agents to treat hyperpigmentation diseases. Zebrafish toxicological model was utilized to further evaluate the toxicity assessment of FLA and FLB in association with the depigmenting effect. Our findings offer the refinement of both compounds as reliable and competent melanogenic inhibitors in zebrafish model. To sum up, both chalcone derivatives of FLA and FLB have the potential to be developed as depigmenting agents for cosmetic or health purposes.

## 4. Materials and Methods

### 4.1. General Procedure for Preparation of Analogues

FLA and FLB were synthesized by Claisen-Schmidt condensation of respective acetophenone and benzaldehyde [40]. An effective straight-forward synthesis of desired chalcone derivatives were illustrated as in Figure 7A. Products were finally purified by using flash column chromatography, with ratio of ethylacetate:hexane of 1:1. FLA and FLB were then treated with MeOH to yield yellow recrystallized compounds. Compounds were characterized based on spectroscopic techniques such as IR, UV, EI-MS and NMR data, consistent as previously reported [40].

#### 4.1.1. (*E*)-1-(2′-Hydroxy-4′,6′-dimethoxyphenyl)-3-(4-methoxyphenylprop-2-en-1-one: (FLA)

Yellow needles crystals. Yield 82.4%. m.p 110–112 °C. (Molecular formula C_18_H_18_O_5_). IR (KBr): 1654 (C=O), 3200 (Ar C-H str.), 1590, 1589, 1476, (C=C), 1202 (C-O str.), 850 cm^−1^. ^1^H-NMR (CDCl_3_, 500 MHz): δ 13.6 (s, 1H, OH), 8.02 (d, 1H, *J* = 15.5 Hz, Hβ), 7.81 (d, 1H, *J* = 15.5 Hz, Hα), 7.73 (brd, 2H, H-2, 6), 7.49 (d, 2H, H-3, 5), 6.87 (br s, 1H, H-5’), 6.02 (br s, 1H, H-3’), 3.93 (s, 3H, OMe, C-6’), 3.85 (s, 3H, OMe, C-4’), 3.82 (s, 3H, OMe, C-4). ^13^C-NMR (CDCl_3_, 125 MHz): δ 192.6 (C=O), 168.7 (C2’), 166.2 (C4’), 163.1 (C6’), 160.9 (C4), 141.6 (Cβ), 130.1(C2, C6), 127.5 (C1), 125.4 (Cα), 114.3 (C3, C5), 93.56 (C3’), 91.4 (C5’), 55.79 (OCH_3_), 55.53 (OCH_3_), 55.35 (OCH_3_). EI-MS *m/z* 298.33.

#### 4.1.2. (*E*)-1-(2′-Hydroxy-4′,6′-dimethoxyphenyl)-3-phenylprop-2-en-1-one:(FLB)

Yellow needles crystals: Yield 82.4%, m.p. 102–104 °C. (Molecular formula C_17_H_16_O_4_). IR (KBr): 3060 (Ar C-H str.), 1642 (C=O), 1589, 1588, 1478, (C=C ring), 1202 (C-O) cm^−1^. ^1^H-NMR (CDCl_3_, 500 MHz): δ 14.3 (s, 1H, OH), 7.92 (d, 1H, *J* = 15.0 Hz, Hβ), 7.80 (d, 1H, *J* = 15.0 Hz, Hα), 7.64 (brd, 2H, H-2, 6), 7.48 (d, 3H, H-3, 4, 5), 6.98 (br s, 1H, H-5’), 6.21 (br s, 1H, H-3’), 3.93 (s, 3H, OMe, C-6’), 3.85 (s, 3H, OMe, C-4’). ^13^C-NMR (CDCl_3_, 125 MHz): δ 193.2 (C=O), 168.9 (C2’), 166.7 (C4’), 162.5 (C6’), 142.51 (Cβ), 141.0 (C4), 132.9 (C1), 130.2 (C2, C6), 128.1 (C3, C5), 126.6 (Cα), 108.14 (C1’), 93.6 (C3’), 90.94 (C5’), 55.53 (OCH_3_), 55.7 (OCH_3_). EI-MS *m/z* 284.23. 

### 4.2. Cell Culture and Treatment

The B16/F10 murine melanoma cells were purchased from the American Type Culture Collection (ATCC, Manassas, VA, USA). The cells were cultured in Dulbecco’s Modified Eagle‘s Medium (DMEM, Sigma-Aldrich, St. Louis, MO, USA) containing 10% fetal bovine serum (FBS) (Gibco, Life Technologies, Waltham, MA, USA), 2 mM l-Glutamine, 4.5 g/L glucose and antibiotics 100 U/mL of penicillin/streptomycin (Gibco, Life Technologies, Carlsbad, CA, USA). The cell line was cultured in 75 cm^2^ flask (TPP, Trasadingen, Switzerland) at 37 °C supplied with 5% carbon dioxide (CO_2_) in fully humidified air. B16/F10 melanoma was adherent cells whereby cells are attached to the surface of flask. The culture medium was changed every alternate day. Cells were harvested by trypsinization using 1× TrypLE Express (Gibco, Life Technologies, Carlsbad, CA, USA). The subcultures were prepared by trypsinization and the sub cultivation ratio was between 1:2 to 1:4 (culture:media). Cells were seeded with appropriate seed density for 24 h and then treated with various concentrations of drugs in the presence of 10 nM α-MSH (melanin inducer). Blank control was set as cells in culture media DMEM only. Cells were harvested after 72 h post-treatment (hpt).

### 4.3. Cell Viability Test

Cell viability test using MTT assay was performed in order to examine the viability of cells upon treatment with different concentrations of samples. This assay depends on the ability of cells to reduce the MTT reagent (3-[4, 5-dimethylthiazol-2-yl]-2, 5-diphenyl tetrazolium bromide) to a purple formazan product. Cells were seeded (1 × 10^5^ cell/well) in 96-well plate (TPP, Trasadingen, Switzerland) for 24 h and replaced with new culture media containing different concentrations of drugs. Mixed media was removed 72 hpt, and replaced with 5 mg/mL MTT reagent in phosphate buffer saline (PBS). Cells containing MTT solution were incubated further for 4 h before the solution was removed and cells were dissolved in 100% DMSO. Dissolved purple formazan crystals were measured at 570 nm in spectrophotometer (SpectraMax, Plus 384, Molecular Devices, Inc., San Jose, CA, USA). Control was set as cells induced with α-MSH and 0.1% DMSO. Cells survival rate were calculated as follows: Cell viability (%) = (A_sample_/A_control_) × 100%.

### 4.4. Determination of Cellular Melanin Content

To determine the ability of samples in attenuating the excessive production of cellular melanin, measurement of melanin content in α-MSH-induced B16/F10 cells was conducted according to Chan et al. [65] with minor modifications. Cells were seeded (2 × 10^5^ cells/well) for 24 h in 6-well culture plates with 3 mL of medium prior to different concentrations of drugs for 72 h. At the end of cell treatment, cells were washed twice with PBS and harvested at 15,000 × *g* for 15 min. The pellets obtained were lyzed with 1 mL of 1 N NaOH in 10% DMSO and incubated at 80 °C for 1 h to dissolve the melanin. Absorbance was read at 475 nm by microplate reader (Sunrise, Tecan, Switzerland). Melanin content was evaluated from standard curve (0.78–100 µg/mL) prepared from authentic standard of synthetic melanin (Sigma, St. Louis, MO, USA).

### 4.5. Determination of Cellular Tyrosinase 

To examine the inhibitory effect of samples on tyrosinase activity, cellular tyrosinase activity of α-MSH-induced B16/F10 was determined by employing the method outlined by Lin et al. [66], with minor modifications. Tyrosinase activity in cells was evaluated by measuring the oxidation rate of l-DOPA. Cells were seeded (2 × 10^5^ cells/well) for 24 h in 6-well culture plates with 3 mL of medium and harvested at 72 hpt. Cells were washed twice with ice-cold PBS then lysed in lysate buffer of 20 mM phosphate buffer (pH 6.8) containing 1 mM PMSF and 1% Trion X-100. Membrane-bound tyrosinase was released from melanosomes with the presence of detergent. Cells were then disrupted by using sonicator with 30 kHz frequency wave in 15 s sonication burst cycles on ice. Lysates were then clarified by centrifugation at 15,000× *g* at 4 °C for 15 min. Protein content in supernatant was determined by using Bradford assay (Bio-Rad Laboratories Inc, Hercules, CA, USA) with bovine serum albumin (BSA, Sigma, USA) used as protein standard. Cellular tyrosinase assay was measured by adding 0.4 µg of cell lysate protein into reaction mixture of 50 mM of phosphate buffer (pH 6.8) and 2.5 mM of l-DOPA. After incubation of 15 min, absorbance readings were monitored at 475 nm by spectrophotometer (Sunrise, Tecan, Switzerland). The amount of dopachrome formation was calculated using the Beer–Lambert Law applying the molar extinction coefficient of dopachrome value 3600 M^−1.^cm^−1^. One unit of tyrosinase activity is equivalent to the total enzyme (tyrosinase) that catalyzes the formation of 1 µmol of dopachrome per min. The specific tyrosinase activity was evaluated by normalizing protein content in the reaction.

### 4.6. Quantitative Real-Time PCR (RT-qPCR) of Tyr, Trp-1, Trp-2 and Mitf

FLA- and FLB-treated of α-MSH-induced B16/F10 melanoma cells were harvested and centrifuged at 300× *g* for 5 min in 4 °C. Total RNA was extracted according to manufacturer’s protocol of RNeasy Plus Mini kit (Qiagen, Hilden, Germany). cDNA was synthesized according to protocol by iScript Reverse Transcription Supermix for RT-qPCR (Bio-Rad Laboratories Inc, Hercules, CA, USA). Real-Time PCR was performed as according to manufacturer’s protocol of iTaq Universal SYBR Green Supermix (Bio-Rad, Hercules, CA, USA). Reaction mixtures included 0.5 µM of both forward and reverse primers, cDNA template (0.1 µg RNA) and 2× iTaq Universal SYBR Green Supermix. Standard curves of each targeting gene were conducted by performing ten-fold serial dilutions of template cDNA. All genes of interest were amplified using similar thermal cycling conditions as follows; 95 °C for 3 min (DNA Polymerase activation) followed by 95 °C for 10 s (DNA denaturation) and 30 s of specific annealing/extension temperatures (*Tyr* at 60 °C, *Trp*-1 at 52 °C, *Trp*-2 at 52 °C, *Mitf* at 60 °C, *ß-actin* at 58 °C and *GAPDH* at 58 °C) with 40 cycles for each gene expression analysis. Real-time PCR analysis was carried out by normalizing expression ratio using Livak method of 2^−(ΔΔCt)^ [67]. Fold differences between C_t_ value of control samples with corresponding gene of interest was calculated as ΔΔC_t_ = ΔC_t_
_treated sample_ − ΔC_t_
_control samples_. Both *ß-actin* and *GAPDH* were used as reference genes in the experiment. All samples were run in triplicate and values represent the mean ± SD of normalized fold difference. Sequences of forward and reverse primers were listed as in Table 1.

### 4.7. Zebrafish Husbandry and Embryos Collection

The present in vivo studies were conducted in obligations to policies of Institutional Animal Care and Use Committee of Universiti Pura Malaysia (IACUC/AUP-R-24/2014). Wild-type shortfin phenotype of adult zebrafish (*Danio rerio*), was purchased from Danio Assay Laboratories Sdn. Bhd. (Malaysia) and bred in-house at Department of Biochemistry, Faculty of Biotechnology and Biomolecular Sciences, UPM. Male and female adult zebrafish at the ratio of 1:2 were let to acclimatized in a recirculating dechlorinated freshwater aquarium system with controlled conditions for a week at 28 ± 1 °C on a 14 h light:10 h dark photoperiod cycle to induce spawning [68]. Adult zebrafish was fed twice daily with brine shrimp (*Artemia salina,* San Francisco Bay Brand, San Francisco, CA, USA) in the morning and flake foods in the evening. The collection container was placed with artificial aquatic plants to imitate spawning sites [69] and fertilized eggs were collected 1 h post light onset [70]. Embryos were washed once with distilled water and twice with embryo media (15 mM NaCl, 0.5 mM KCl, 1 mM MgSO_4_, 0.15 mM KH_2_PO_4_, 0.05 mM Na_2_HPO_4_, 1 mM CaCl_2_ and 0.7 mM NaHCO_3_, pH 7.0). Successfully fertilized embryos with intact chorion membranes and reached gastrulation stage 50% epiboly at approximately 5 to 9 hpf [71] were selected under dissecting microscope (Leica Zoom 2000) at magnification of 40×. 

### 4.8. Zebrafish Toxicity Assay of FLA and FLB

The toxicity effects of FLA and FLB during the embryonic stage of zebrafish were evaluated on parameters such as spontaneous tail coiling (24 hpf), heartbeat activity (48 hpf), mortality (72 hpf) and morphological deformities throughout the exposure time [28,29]. Twelve fertilized eggs at 9 hpf were selected for each concentration tested and control (*n* = 96). For the calculation of LC50, the OECD guidelines for Fish Embryo Acute Toxicity Test (TG 236) were used [72]. Dead embryos were identified and removed immediately to avoid any contamination during exposure time when it forms coagulation, non-detachment of tail, lack of somite formation as well as absence of heartbeat [72]. Mortality and teratogenic effect of drugs on zebrafish embryos were observed daily between 48 to 144 hpf by using inverted microscope (Nikon Eclipse TS 100 Shinagawa, Japan). The number of zebrafish embryo’s tail coiling was counted within 1 min period. Cardiac toxicity was evaluated by measuring heart rates of zebrafish embryos in natural lateral recumbent position within 1 min period without anesthesia to avoid bias heartbeat scoring [73,74]. Zebrafish larvae were not fed during the experiment. 

### 4.9. Phenotype-Based Depigmenting Test of FLA and FLB on Zebrafish

Phenotype-based melanogenesis inhibition of zebrafish embryos treated with FLA and FLB were determined following reported method [75]. To observe depigmenting of FLA and FLB on zebrafish, fertilized embryos at 9 hpf were selected and arrayed in 96 well-plate with single embryo per well. Control was set as embryos media containing 0.1% DMSO. Embryos treated with IBMX at 100 μM was set as negative control. Positive control was set as embryos maintained in 200 µL of embryo media containing 100 μM of PTU from 9 to 96 hpf. Treatment of the zebrafish with FLA and FLB was carried out at 96 hpf after PTU was washed off and embryos were treated simultaneously with 100 μM IBMX to induce melanogenesis. PTU was used to suppress melanogenesis in zebrafish without adverse toxicity [76,77]. IBMX was used as better melanogenic inducer agent in zebrafish model instead of α-MSH to mimic cAMP signaling pathway in melanogenesis omitting any effects of α-MSH towards zebrafish [30,78]. Inhibition of melanin by FLA and FLB in 0.1% DMSO was observed every 48 h until 144 hpf. The developmental phenotypes of the experimental embryos were observed every 24 h for 144 h by using inverted microscope Nikon Eclipse TS100 and were imaged.

### 4.10. Statistical Analysis

All the results were presented as the mean ± standard error of mean (S.E.M) of three independent experiments. Differences between groups were determined by using one-way analysis of variance (ANOVA) followed by Dunnett test. Values of * *p* < 0.05, ** *p* < 0.01 and *** *p* < 0.001 were considered significant. All graphs were generated by using GraphPad Prism version 5.0 (GraphPad Software, Inc., San Diego, CA, USA).

## Figures and Tables

**Figure 1 molecules-25-03403-f001:**
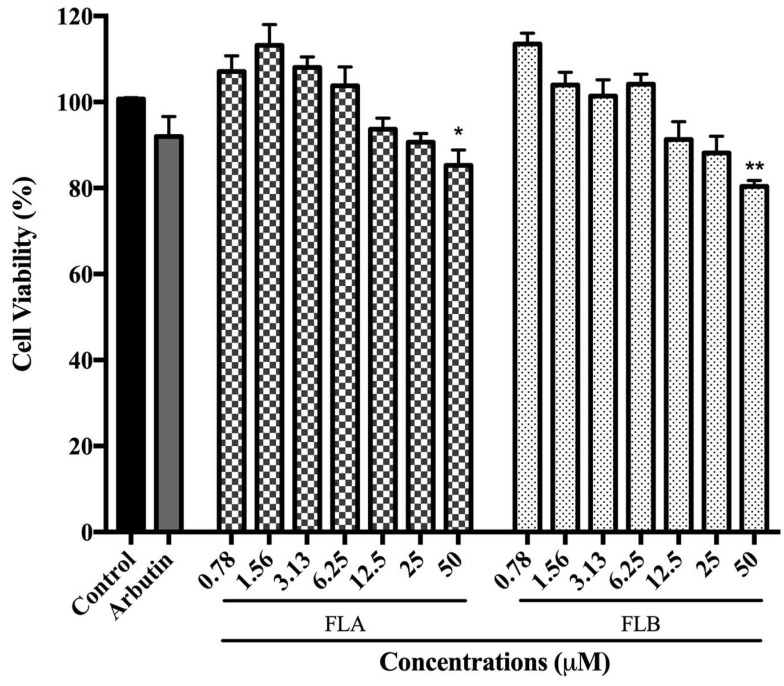
Cell viability of α-MSH induced B16/F10 cells after 72 h treatment with FLA and FLB. Toxicity effects of FLA and FLB were determined with respect to induced control by MTT assay. FLA and FLB were tested for cytotoxic effect at different concentrations of 0.78–50 µM. Arbutin (50 µM) was used as positive control. All values are the mean ± S.E.M. of three independent experiments. * *p* < 0.05 and ** *p* < 0.01 were considered significantly different to α-MSH-induced control group.

**Figure 2 molecules-25-03403-f002:**
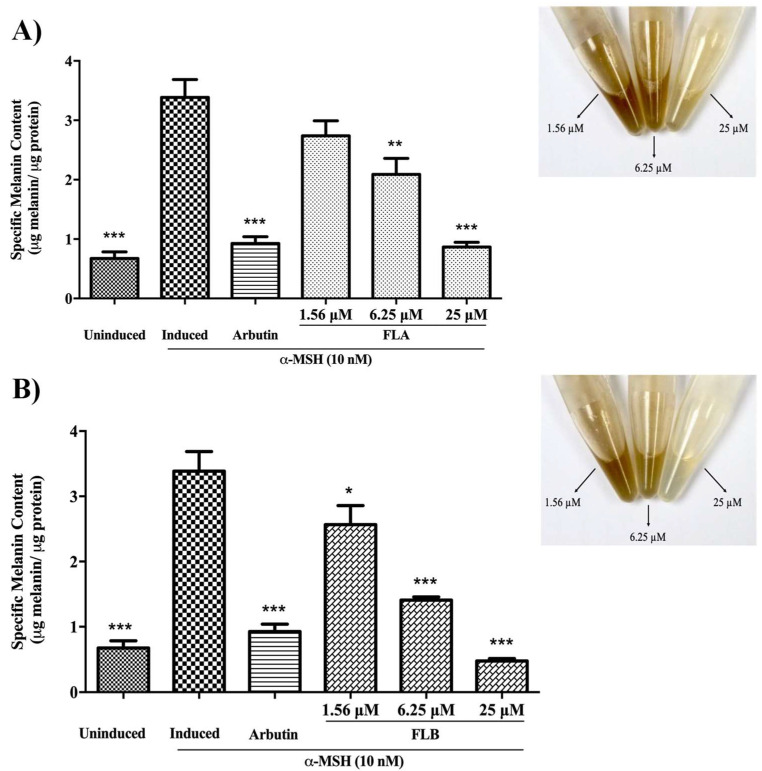
Effect of chalcone derivatives on melanin production of α-MSH-induced B16/F10 cells. Cellular melanin content after treated with (**A**) FLA and (**B**) FLB. Both compounds were tested at different concentrations (25, 6.25 and 1.56 µM) and incubated for 72 h. Arbutin (50 µM) was used as positive control. All values are the mean ± S.E.M. of three independent experiments. ** *p* < 0.01 and *** *p* < 0.01 were considered significantly different to α-MSH-induced control group.

**Figure 3 molecules-25-03403-f003:**
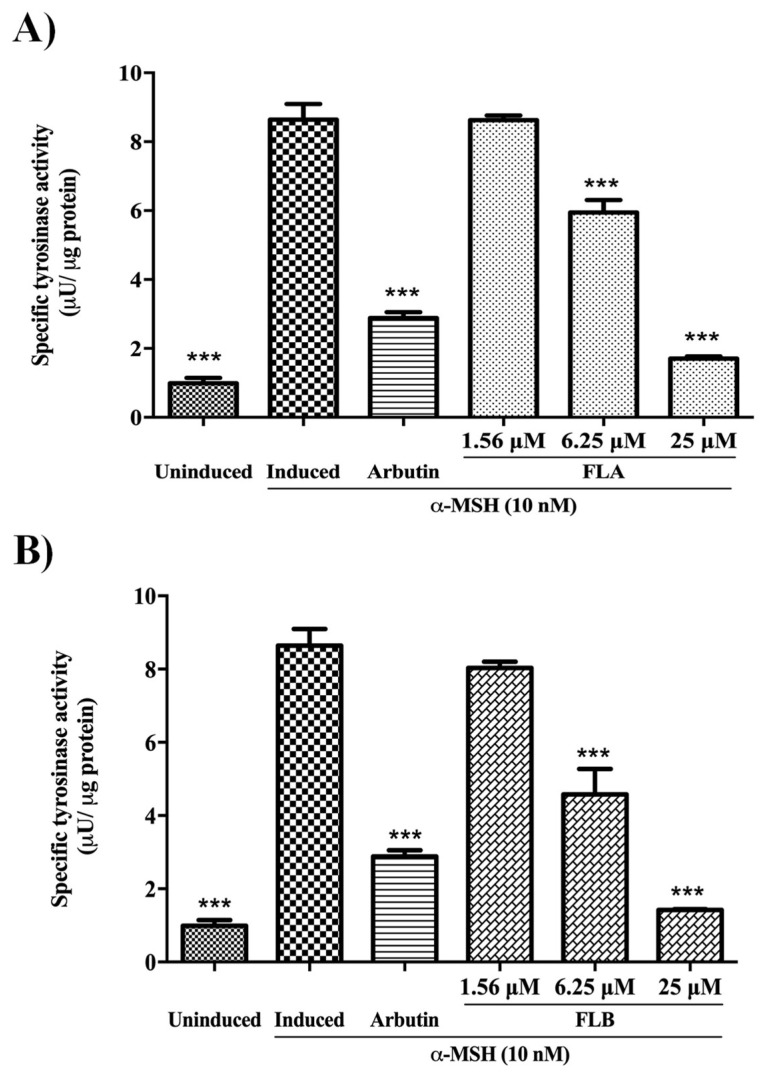
Effect of chalcone derivatives on cellular tyrosinase activity of α-MSH-induced B16/F10 cells. Cellular tyrosinase activity after treated with (**A**) FLA and (**B**) FLB. Both compounds were tested at different concentrations (25, 6.25 and 1.56 µM) and incubated for 72 h. Arbutin (50 µM) was used as positive control. All values are the mean ± S.E.M. of three independent experiments. A value of *** *p* < 0.001, was considered significantly different to α-MSH-induced control group.

**Figure 4 molecules-25-03403-f004:**
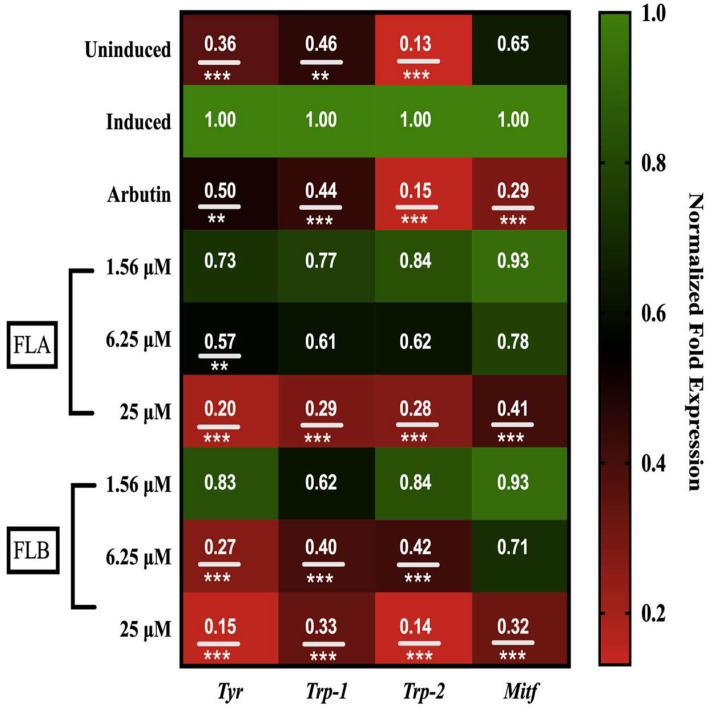
Effect of chalcone derivatives on the mRNA expression of melanogenic-related genes: tyrosinase (*Tyr*), tyrosine-related protein 1 and 2 (*Trp*-1 and -2) and microphthalmia-associated transcription factor (*Mitf*). B16/F10 cells (2 × 10^5^ cells/well) were treated with 10 nM of α-MSH for 72 h in the presence or absence of FLA and FLB. Total RNA was extracted, and mRNA expression was analyzed by RT-qPCR. The fold expression results shown in figure were normalized by *GAPDH* and *ß-actin* mRNA levels. Data are expressed as the mean ± SEM of three separate experiments. A value of ** *p* < 0.01 and *** *p* < 0.001 were considered significantly different to α-MSH-induced control group.

**Figure 5 molecules-25-03403-f005:**
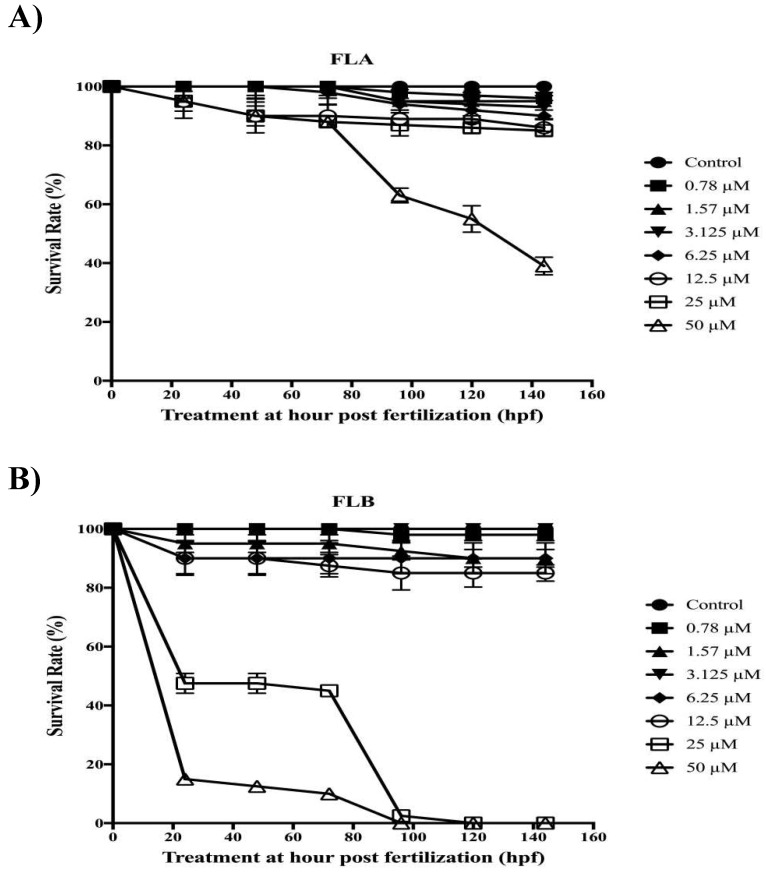
Toxicity effect of chalcone derivatives on zebrafish embryos. Embryos were incubated in seven different concentrations (0.78 μM–50 μM) of (**A**) FLA and (**B**) FLB. Control was set as embryos treated in embryo media with 0.1% DMSO. The survival rates were recorded every 24 h until 144 h by using inverted microscope (Nikon TS100). Twelve fertilized eggs at 9 h post fertilization were selected for each concentration tested and control. Each value in the table is represented as mean ± S.E.M (*n* = 4).

**Figure 6 molecules-25-03403-f006:**
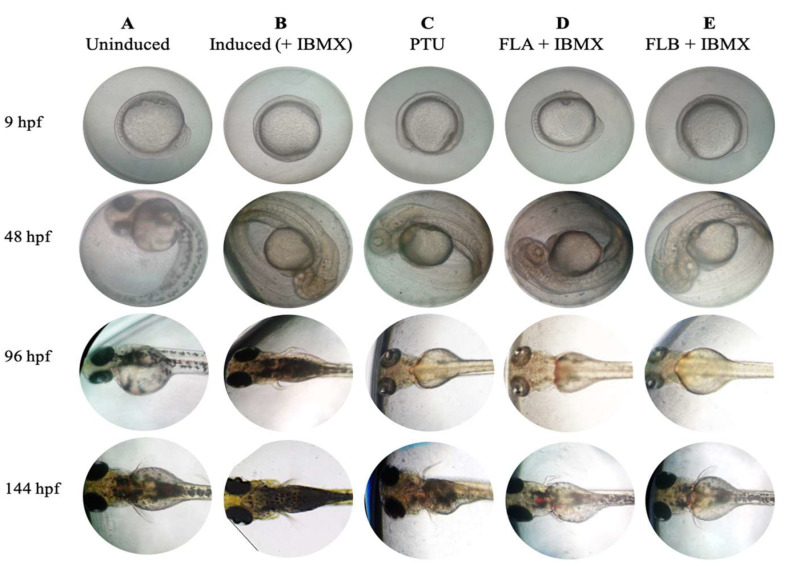
Phenotype-based melanogenesis inhibition of chalcone derivatives on zebrafish model. Synchronized zebrafish embryos were collected and maintained with (**A**) embryo media with 0.1% DMSO (uninduced) and (**B**) IBMX (100 μM) in embryo media with 0.1% DMSO (induced). Zebrafish embryos (**C**–**E**) were treated with PTU (200 µM) from 9–96 hpf. Treatment of (**D**) FLA at 25 µM and (**E**) FLB at 6.25 µM was carried out at 96 hpf after PTU-washed off and treated simultaneously with IBMX at 100 µM.

**Figure 7 molecules-25-03403-f007:**
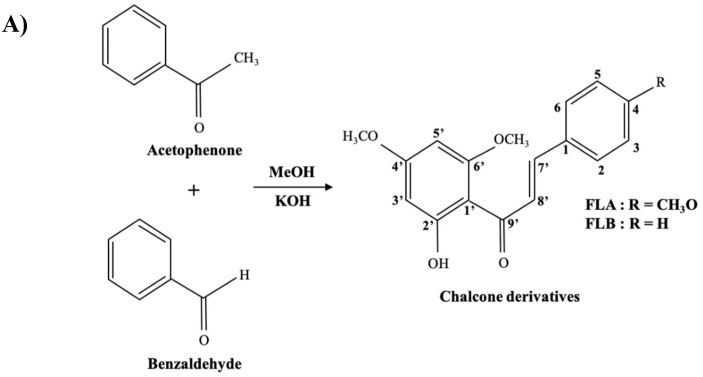
Synthesis of chalcone derivatives and chemical structures of compounds. (**A**) Synthesis of Flavokawain A (FLA) and flavokawain B (FLB). (**B**) Skeleton structure of chalcone resembles the skeleton structures of tyrosinase substrates which are tyrosine and l-DOPA.

**Table 1 molecules-25-03403-t001:** Sequences of *Tyr*, *Trp*-1, *Trp*-2 and *Mitf* used in RT-qPCR.

Gene	Primer Sequences	Accession Number
*Tyr*	Forward: TTG CCA CTT CAT GTC ATC ATA GAA TAT T Reverse: TTT ATC AAA GGT GTG ACT GCT ATA CAA AT	NM_011661.5
*Trp*-1	Forward: GCT GCA GGA GCC TTC TTT CTC Reverse: AAG ACG CTG CAC TGC TGG TCT	NM_031202.3
*Trp*-2	Forward: GGA TGA CCG TGA GCA ATG GCC Reverse: CGG TTG TGA CCA ATG GGT GCC	NM_010024.3
*Mitf*	Forward: TAC AGA AAG TAG AGG GAG GAG GAC TAA G Reverse: CAC AGT TGG AGT TAA GAG TGA GCA TAG CC	NM_008601.3
*GAPDH*	Forward: ACC ACA GTC CAT GCC ATC AC Reverse: TCC ACC ACC CTG TTG CTG TA	NM_008084.3
*ß-actin*	Forward: ACC GTG AAA AGA TGA CCC AG Reverse: TAC GGA TGT CAA CGT CAC AC	NC_007393.5

*Tyr*, Tyrosinase; *Trp*-1 and -2, Tyrosinase related protein 1 and 2; *Mitf*, Microphthalmia-associated transcription factor; *GAPDH*, Glyceraldehyde 3-phosphate dehydrogenase.

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
