# Peer review of "Melanogenic Inhibition and Toxicity Assessment of Flavokawain A and B on B16/F10 Melanoma Cells and Zebrafish (*Danio rerio*)"

_molecules, 2020, doi:10.3390/molecules25153403_

Round 1

Reviewer 1 Report

In this manuscript, the toxicity and melanogenic inhibition activity of two synthetic compounds (Flavokawain A and Flavokawain B) were studied with the model of B16/F10 melanoma cells and zebrafish. Flavokawain A and Flavokawain B showed no toxicity effect in B16/F10 cells and zebrafish at 25 µM and 6.25 µM, respectively. The outstanding melanogenic inhibition activity may be related to the suppressing the expression of Tyr, Trp-1, Trp-2 and Mitf genes. This is a well-written paper with good experimental design. In general, the article deserves publication after some suitable revisions.

  1. Please give the correct order of all Figures according to the content of the manuscript.
  2. The chemical structures in Figure 1 and Figure 8 can be given in one figure.
  3. In Figure 3 and 4, “Both compounds were tested at different concentrations (2 µM, 10 µM and 50 µM)”, however, “1.56 µM, 6.25 µM and 25 µM” were tested according to the manuscript.
  4. It is recommended to add the test of the expression of melanogenic-related genes in zebrafish.

Author Response

RESPONSE TO REVIEWER 1 COMMENTS.

Point 1: Please give the correct order of all Figures according to the content of the manuscript.

- The order of the figures was placed incorrectly during submission. All figures have been re-arranged in the correct order according to the content of the manuscript.

Point 2: The chemical structures in Figure 1 and Figure 8 can be given in one figure.

- The order of the figures was placed incorrectly during submission. Figure 1 which was supposedly listed as Figure 7 (skeleton structures of chalcone, tyrosine and L-DOPA) in the manuscript has been combined with Figure 8 (synthesis of chalcone derivatives). The new Figure is labeled as Figure 7.

Point 3: In Figure 3 and 4, “Both compounds were tested at different concentrations (2 µM, 10 µM and 50 µM)”, however, “1.56 µM, 6.25 µM and 25 µM” were tested according to the manuscript.

- The order of the figures was placed incorrectly during submission. Figures 3 and 4 in the manuscript were re-ordered and become Figures 2 and 3, respectively. The concentrations tested were 1.56 µM, 6.25 µM and 25 µM. Corrections have been made in the legends of Figures 2 and 3.

Point 4: It is recommended to add the test of the expression of melanogenic-related genes in zebrafish.

- The expression of melanogenic-related genes in zebrafish was not tested. Due to a few limitations, the current study only focuses on phenotype-based of melanogenic inhibition and toxicity assessment in the zebrafish model which provides important key information of the depigmenting properties of FLA and FLB.

Reviewer 2 Report

Melanogenic Inhibition and Toxicity Assessment of Flavokawain A and B on B16/F10 Melanoma Cells and Zebrafish (Danio rerio)

The manuscript is interesting, however, as it is presented, it is not possible to evaluate, because, it has serious flaws.

The work should be sent to the authors, for their review and ordering of the sections with their respective results.

After these important changes, the work should be resubmitted for evaluation

There are figures that should be removed (1), and the location of others should be modified (2-7).

Some of the sections that should be reviewed are given below.

Lines 26-2. The paragraph should be revised and improved

“Our findings provide elaborated phenotype-based depigmenting activity and toxicity assessment of FLA and FLB and on zebrafish model to rule them out as potent depigmenting agents. Both chalcone derivativescould be used as lead compounds to develop new anti-pigmenting agents.”

Section 2.1. Cytotoxic effect of chalcone derivatives on B16/F10 cells

Lines 88-89, page 2

“The toxicity effects of FLA and FLB on α-MSH-induced B16/F10 cells are illustrated as in Figure 1.”

However, in figure 1 is showed

Figure 1. of chalcone derivatives: Flavokawain A (FLA) and Flavokawain B (FLB).

Sections 2.2. Inhibitory effect of chalcone derivatives on cellular melanin content

The authors write in lines 101-102 page 3;”The depigmenting  effect of both FLA and FLB on α-MSH-induced B16/F10 cells is described as in Figure 2.”

However, in Figure 2 are showed

Figure 2. Cell viability of α-MSH induced B16/F10 cells after 72 h treatment with FLA and FLB.

Section 2.3. Inhibitory effect of chalcone derivatives on cellular tyrosinase activity

The authors write in lines 121-122, page 4: “The melanogenic effect of chalcone derivatives was evaluated on cellular tyrosinase activity of α- MSH-induced B16/F10 cells (Figure 3).”

However, in Figure 3 are showed

Figure 3. of chalcone derivatives on melanin content production of α-MSH induced B16/F10 cells.

Section 2.4. Down-regulation effect of chalcone derivatives on Tyr, Trp-1, Trp-2 and Mitf genes expression in B16/F10 cells.

The authors write in lines 138-140, page 5:

“Mitf as well as Tyr, Trp-1, and Trp-2 genes are well-studied to regulate majorly in melanogenesis.The effect of FLA and FLB on mechanism of melanin biosynthesis pathway was elucidated on Tyr,Trp-1, Trp-2 and Mitf genes expression (Figure 4).

However, in Figure 4 are showed:

Figure 4. of chalcone derivatives on cellular tyrosinase activity of α-MSH induced B16/F10 cells.

  • Section 2.5. Zebrafish toxicity assessment

Lines 162-164, page 6: “Figure 5 showed the survival rates of zebrafish post-treated with 10-fold serially diluted FLA and FLB (0.78-50 μM) in 0.1% DMSO with the presence of IBMX inducer at 100 μM.”

However, in Figure 5 are showed:

Figure 5. of chalcone derivatives on the mRNA expression of melanogenic-related genes: tyrosinase (Tyr), tyrosine-related protein 1 and 2 (Trp-1 and -2) and microphthalmia-associated transcription factor (Mitf)

In Figures 3-5, please revise

Figure 5. of chalcone. Apparently by typing error a word was omitted before “of”.

The work is not suggested for publication. 

Author Response

RESPONSE TO REVIEWER 2 COMMENTS.

Point 1: Lines 26-2. The paragraph should be revised and improved.

“Our findings provide elaborated phenotype-based depigmenting activity and toxicity assessment of FLA and FLB and on zebrafish model to rule them out as potent depigmenting agents. Both chalcone derivatives could be used as lead compounds to develop new anti-pigmenting agents.”

- The lines were revised and improved accordingly. (page 1, line 26-30). "Further experiments on the zebrafish model demonstrated successful phenotype-based depigmenting activity of FLA and FLB under induced melanogenesis. To sum up, our findings provide important first key step for both of the chalcone derivatives to be further studied and developed as potent depigmenting agents."

Point 2: Lines 88-89, page 2

“The toxicity effects of FLA and FLB on α-MSH-induced B16/F10 cells are illustrated as in Figure 1.” However in figure 1 is showed Figure 1. of chalcone derivatives: Flavokawain A (FLA) and Flavokawain B (FLB).

- The order of the figures was placed incorrectly during submission. Thus, all figures have been re-ordered. Figure 1 in the manuscript is corrected as follows;

Figure 1   Cell viability of α-MSH induced B16/F10 cells after 72 h treatment with FLA and FLB. Toxicity effects of FLA and FLB were determined with respect to induced control by MTT assay. FLA and FLB were tested for cytotoxic effect at different concentrations of 0.78-50 µM. Arbutin (50 µM) was used as positive control. All values are the mean ± S.E.M. of three independent experiments. *P<0.05 and **P<0.01 were considered significantly different to the α-MSH-induced control group.

Point 3: Sections 2.2. Inhibitory effect of chalcone derivatives on cellular melanin content

The authors write in lines 101-102 page 3;”The depigmenting  effect of both FLA and FLB on α-MSH-induced B16/F10 cells is described as in Figure 2.”However in Figure 2 are showed Figure 2. Cell viability of α-MSH induced B16/F10 cells after 72 h treatment with FLA and FLB.

- The order of the figures was placed incorrectly during submission. Thus, all figures have been re-ordered. Figure 2 in the manuscript is corrected as follows;

Figure 2   Effect of chalcone derivatives on melanin content production of α-MSH induced B16/F10 cells. Cellular melanin content after treated with A) FLA and B) FLB. Both compounds were tested at different concentrations (25, 6.25 and 1.56 µM) and incubated for 72 h. Arbutin (50 µM) was used as positive control. All values are the mean ± S.E.M. of three independent experiments. *P<0.05, **P<0.01 and ***P<0.01 were considered significantly different to the α-MSH-induced control group.

Point 4: Section 2.3. Inhibitory effect of chalcone derivatives on cellular tyrosinase activity

The authors write in lines 121-122, page 4: “The melanogenic effect of chalcone derivatives was evaluated on cellular tyrosinase activity of α- MSH-induced B16/F10 cells (Figure 3).” However in Figure 3 are showed Figure 3. of chalcone derivatives on melanin content production of α-MSH induced B16/F10 cells.

- The order of the figures was placed incorrectly during submission. Thus, all figures have been re-ordered. Figure 3 in the manuscript is corrected as follows;

Figure 3   Effect of chalcone derivatives on cellular tyrosinase activity of α-MSH induced B16/F10 cells.  Cellular tyrosinase activity after treated with A) FLA and B) FLB. Both compounds were tested at different concentrations (25, 6.25 and 1.56 µM) and incubated for 72 h. Arbutin (50 µM) was used as positive control. All values are the mean ± S.E.M. of three independent experiments. A value of ***P<0.001, was considered significantly different to α-MSH-induced control group.

Point 5: Section 2.4. Down-regulation effect of chalcone derivatives on Tyr, Trp-1, Trp-2 and Mitf genes expression in B16/F10 cells. The authors write in lines 138-140, page 5: “Mitf as well as Tyr, Trp-1, and Trp-2 genes are well-studied to regulate majorly in melanogenesis.The effect of FLA and FLB on mechanism of melanin biosynthesis pathway was elucidated on Tyr,Trp-1, Trp-2 and Mitf genes expression (Figure 4). However in Figure 4 are showed: Figure 4. of chalcone derivatives on cellular tyrosinase activity of α-MSH induced B16/F10 cells.

- The order of the figures was placed incorrectly during submission. Thus, all figures have been re-ordered. Figure 4 in the manuscript is corrected as follows;

Figure 4   Effect of chalcone derivatives on the mRNA expression of melanogenic-related genes: tyrosinase (Tyr), tyrosine-related protein 1 and 2 (Trp-1 and -2) and microphthalmia-associated transcription factor (Mitf). B16/F10 cells (2x105 cells/well) were treated with 10 nM of α-MSH for 72 h in the presence or absence of FLA and FLB. Total RNA was extracted, and mRNA expression was analyzed by RT-qPCR. The fold expression results shown in A) and B)were normalized by GAPDH and ß-actin mRNA levels. Data are expressed as the mean ± SEM of three separate experiments. A value of **P<0.01 and ***P<0.001 were considered significantly different to the α-MSH-induced control group.

Point 6: Section 2.5. Zebrafish toxicity assessment

Lines 162-164, page 6: “Figure 5 showed the survival rates of zebrafish post-treated with 10-fold serially diluted FLA and FLB (0.78-50 μM) in 0.1% DMSO with the presence of IBMX inducer at 100 μM.”However in Figure 5 are showed: Figure 5. of chalcone derivatives on the mRNA expression of melanogenic-related genes: tyrosinase (Tyr), tyrosine-related protein 1 and 2 (Trp-1 and -2) and microphthalmia-associated transcription factor (Mitf)

- The order of the figures was placed incorrectly during submission. Thus, all figures have been re-ordered. Figure 5 in the manuscript is corrected as follows;

Figure 5   Toxicity effect of chalcone derivatives on zebrafish embryos. Embryos were incubated in seven different concentrations (0.78 μM-50 μM) of A) FLA and B) FLB. Control was set as embryos treated in embryo media with 0.1% DMSO. The survival rates were recorded every 24 hours until 144 hours by using an inverted microscope (Nikon TS100). Twelve fertilized eggs at 9 hours post-fertilization were selected for each concentration tested and control. Each value in the table is represented as mean ± S.E.M (n=4).

Point 7: In Figures 3-5, please revise. Figure 5. of chalcone. Apparently by typing error a word was omitted before “of”.

- The order of the figures was placed incorrectly during submission. Thus, all figures have been re-ordered. Figure 5 in the manuscript was re-ordered and becomes Figure 4. The omitted word(s) before “of” in Figures 3-5 was added.

Reviewer 3 Report

This work is in follow up to previous papers where they showed the inhibitory activity of chalcone derivatives against tyrosinase (J. Mol. Struc., 2015,1085, 97-103). In this manuscript, the authors reported the anti-melanogenic activity and toxicity of flavokawain A and B on B16/F10 melanoma cells and zebrafish. Although the experimental results support the author's conclusions, this manuscript has a problem of novelty and many lacks as following:

  1. Authors didn’t fully describe previous studies about the anti-melanogenic activity of flavokawain A and B. Jeong et al. already reported the anti-melanogenic activity of flavokawain A and B using MSH-activated B16 melanoma cells (Bioorg. Med. Chem. Lett., 2015, 25, 799-802). Even they studied under the same cell line and method. This point diminishes the novelty and significance of the present study. The authors have to remove the overlapping part or describe the difference compared to the previous paper.
  2. In page 2, line 76, flavokawain A (FLA) and flavokawain B (FLB) are not kavalactones. Please rewrite.
  3. Authors should check and rewrite figure legends for Figures 1, 3, 4, and 5.
  4. The authors gave wrong numbers for all figures in the text. Please change the figure numbers carefully.
  5. If authors didn’t prepare a Supplementary Materials, please remove lines 422 and 423, page 14.
  6. Please give numbering for a chemical structure in Figure 1.
  7. In the line 291, page 11, m of 7.49 (m, 2H, H-3, 5) is should be doublet (d). Please check NMR spectrum and give J value.
  8. In the lines 291 and 296, page 11, br, s, --> br s,
  9. In the line 334, page 12, please remove “assay” from the subtitle “4.5. Determination of cellular tyrosinase assay”

Author Response

RESPONSE TO REVIEWER 3 COMMENTS. 

Point 1: Authors didn’t fully describe previous studies about the anti-melanogenic activity of flavokawain A and B. Jeong et al. already reported the anti-melanogenic activity of flavokawain A and B using MSH-activated B16 melanoma cells (Bioorg. Med. Chem. Lett., 2015, 25, 799-802). Even they studied under the same cell line and method. This point diminishes the novelty and significance of the present study. The authors have to remove the overlapping part or describe the difference compared to the previous paper.

- Jeong et al. (2015) has previously reported the melanogenesis inhibition activity of FLB and FLC, but the FLA was found to exhibit no melanin inhibitory activity. Although the cellular melanin content was tested similarly in MSH-activated B16 melanoma cells, the authors did not determine the cellular tyrosinase inhibition activity of FLA and FLB. Our findings were able to correlate the successful cellular tyrosinase inhibition with the reduction of cellular melanin production. We further elucidated the inhibitory effect of FLA and FLB on the mechanism of melanin biosynthesis pathway through validation of Mitf expression which is the master regulator as well as the other melanogenic-related genes including Tyr, Trp-1 and Trp-2.

Point 2: In page 2, line 76, flavokawain A (FLA) and flavokawain B (FLB) are not kavalactones. Please rewrite.

- The sentence kavalactones have been replaced with kava chalcones.

Point 3: Authors should check and rewrite figure legends for Figures 1, 3, 4, and 5.

The order of the figures was placed incorrectly during submission. Thus, all figures have been re-ordered. Figure 1, 3, 4 and 5 in the manuscript is re-ordered and becomes Figure 7, 2, 3 and 4, respectively. The figure legends have been revised accordingly.

Point 4: Authors gave wrong numbers for all figures in the text. Please change the figure numbers carefully.

- The order of the figures was placed incorrectly during submission. Thus, all figures have been re-ordered carefully.

Point 5: If the authors didn’t prepare a Supplementary Materials, please remove lines 422 and 423, page 14.

- Supplementary Materials 1 and 2 are available but might have been mistakenly omitted during submission.

Point 6: Please give numbering for a chemical structure in Figure 1.

- Figure 1 has been re-ordered and becomes Figure 7A. The chemical structure has been numbered.

Point 7: In the line 291, page 11, m of 7.49 (m, 2H, H-3, 5) is should be doublet (d). Please check NMR spectrum and give J value. 

- The m of 7.49 (m, 2H, H-3, 5) has been corrected into 7.49 (d, 2H, H-3, 5). NMR spectrum and J value have been added which is consistent as previously reported (Akhtar et al. [40]).

Point 8: In the lines 291 and 296, page 11, br, s, --> br s,

- The (br,s,) has been corrected into (br s) in both lines.

Point 9: In the line 334, page 12, please remove “assay” from the subtitle “4.5. Determination of cellular tyrosinase assay”

- The subtitle of 4.5. Determination of cellular tyrosinase assay, has been corrected by removing the word ‘assay’.

Round 2

Reviewer 2 Report

The authors have successfully made the major changes requested. The paper should be accepted in its current state.

Author Response

Reviewer 2: The authors have successfully made the major changes requested. The paper should be accepted in its current state.

- Thank you.

Reviewer 3 Report

I read carefully this revised manuscript entitled “Melanogenic Inhibition and Toxicity Assessment of Flavokawain A and B on B16/F10 Melanoma Cells and Zebrafish (Danio rerio)”. Although the authors made some corrections for the revised manuscript, the correction was not enough. They didn’t describe the previous study (Bioorg. Med. Chem. Lett., 2015, 25, 799-802) that showed the anti-melanogenic activity of flavokawain A and B using MSH-activated B16 melanoma cells. The authors have to describe the difference compared to the previous paper in the text. Thus, I can’t fully accept the author’s responses.

More minor points:

For Figure 7, please switch the figures A) and B) because A) and B) didn’t match with their figure legend.

For reference 31, please keep reference notation.

Author Response

Point 1: I read carefully this revised manuscript entitled “Melanogenic Inhibition and Toxicity Assessment of Flavokawain A and B on B16/F10 Melanoma Cells and Zebrafish (Danio rerio)”. Although the authors made some corrections for the revised manuscript, the correction was not enough. They didn’t describe the previous study (Bioorg. Med. Chem. Lett., 2015, 25, 799-802) that showed the anti-melanogenic activity of flavokawain A and B using MSH-activated B16 melanoma cells. The authors have to describe the difference compared to the previous paper in the text. Thus, I can’t fully accept the author’s responses.

- We added a previous study of Jeong et al (2015) (Bioorg. Med. Chem. Lett., 2015, 25, 799-802) and discussed the difference between our study with the previous report in the manuscript as requested. The correction has been made in the Discussion section.

Point 2: For Figure 7, please switch the figures A) and B) because A) and B) didn’t match with their figure legend.

- The figure legend of A) and B) has been re-ordered to match with the Figure 7. Figure 7 has not been switched to follow the order of Figures 7A and 7B as mentioned in the text.

Point 3: For reference 31, please keep reference notation.

 - The reference 31 notation has been corrected accordingly.